

# Temperature alters the shape of predator–prey cycles through effects on underlying mechanisms

John P. DeLong and Shelby Lyon

School of Biological Sciences, University of Nebraska—Lincoln, Lincoln, NE, USA

## ABSTRACT

**Background:** Predicting the effects of climate warming on the dynamics of ecological systems requires understanding how temperature influences birth rates, death rates and the strength of species interactions. The temperature dependance of these processes—which are the underlying mechanisms of ecological dynamics—is often thought to be exponential or unimodal, generally supported by short-term experiments. However, ecological dynamics unfold over many generations. Our goal was to empirically document shifts in predator–prey cycles over the full range of temperatures that can possibly support a predator–prey system and then to uncover the effect of temperature on the underlying mechanisms driving those changes.

**Methods:** We measured the population dynamics of the *Didinium-Paramecium* predator–prey system across a wide range of temperatures to reveal systematic changes in the dynamics of the system. We then used ordinary differential equation fitting to estimate parameters of a model describing the dynamics, and used these estimates to assess the long-term temperature dependance of all the underlying mechanisms.

**Results:** We found that predator–prey cycles shrank in state space from colder to hotter temperatures and that both cycle period and amplitude varied with temperature. Model parameters showed mostly unimodal responses to temperature, with one parameter (predator mortality) increasing monotonically with temperature and one parameter (predator conversion efficiency) invariant with temperature. Our results indicate that temperature can have profound, systematic effects on ecological dynamics, and these can arise through diverse and simultaneous changes in multiple underlying mechanisms. Predicting the effects of temperature on ecological dynamics may require additional investigation into how the underlying drivers of population dynamics respond to temperature beyond a short-term, acute response.

Corresponding author
John P. DeLong, jpdelong@unl.edu

## INTRODUCTION

Changes in environmental temperature have the potential to alter the dynamics and function of natural systems. With natural or human induced increases in temperature, biological rates including photosynthesis, respiration, movement and reproduction

typically increase, at least up to a point (*Kleiber, 1961*; *Gilloly et al., 2001*; *Brown et al., 2004*). These increases can have cascading effects throughout ecological communities by altering both individual phenotypes and the interactions that occur among individuals and species (*O'Connor, 2009*; *Petchey, Brose & Rall, 2010*; *O'Connor, Gilbert & Brown, 2011*; *Dell, Pawar & Savage, 2014*; *Gilbert et al., 2014*; *Osmond et al., 2017*). Although temperature clearly influences the biochemical processes operating within organisms (*Johnson, 1962*; *Feller, 2010*), how temperature simultaneously influences the full range of processes that govern the dynamics of complex ecological systems is far from understood.

Consumer–resource interactions are sensitive to temperature changes because both foraging rate per se and the strength of the consumer–resource interaction are driven by multiple temperature-dependent processes (*Dell, Pawar & Savage, 2014*; *Gilbert et al., 2014*; *Allan et al., 2015*). This temperature dependance has strong implications for community dynamics and function because of the relationships among interaction strengths, population and community stability (e.g., the occurrence and shape of population cycles), and the overall structure of food webs (*O'Connor et al., 2009*; *Petchey, Brose & Rall, 2010*; *McCann, 2011*; *Binzer et al., 2012*; *Gilbert et al., 2014*; *Gibert et al., 2016*; *Gibert & DeLong, 2017*). Theory predicts a variety of changes in stability and dynamical behavior of consumer–resource communities with changes in temperature, depending upon assumptions about which processes are temperature-dependent and the nature of that dependance (e.g., monotonically increasing or unimodal) (*Vasseur & McCann, 2005*; *Ohlberger et al., 2011*; *O'Connor, Gilbert & Brown, 2011*; *Binzer et al., 2012*; *Fussmann et al., 2014*; *Amarasekare, 2015*; *Gibert et al., 2016*; *Osmond et al., 2017*).

One often overlooked aspect of current theory predicting the effect of warming on community dynamics is the mismatch between the temperature-dependence of rates and the time-scale of the community dynamics. Most models assume that model parameters are a function of temperature, using some function that describes a measured short-term effect of temperature on biological rates. For example, the Boltzmann–Arrhenius function is often used to invoke the kinetic effect of temperature on vital rates or interactions among species (*Vasseur & McCann, 2005*; *Petchey, Brose & Rall, 2010*; *O'Connor, Gilbert & Brown, 2011*; *Fussmann et al., 2014*; *Gilbert et al., 2014*; *Osmond et al., 2017*; *Wang et al., 2019*). This temperature-dependence is then assumed to persist in the same quantitative and qualitative way across many generations. However, when organisms experience a new temperature, they may acclimate or show cross-generational plasticity that alters their response to temperature (*Alexander & McMahon, 2004*; *Osmond et al., 2017*; *Luhring & DeLong, 2017*). Thus, models parameterized with short-term thermal responses can produce mixed results when compared with an observed long-term response (*Yang et al., 2016*). Thus, the emergence of community dynamics at different temperatures is likely to reflect a population's response to temperature post-acclimation, when the temperature dependance of biological processes may have changed.

In response to increasing temperature, populations may show changes in the amplitude or period of cycles (*Nelson, Bjørnstad & Yamanaka, 2013*; *Meisner, Harmon & Ives, 2014*; *Salt et al., 2017*) or the persistence or stability of the system (*Beisner, McCauley & Wrona, 1997*; *Jiang & Kulczycki, 2004*; *Salt et al., 2017*). These patterns of change appear

to be driven by shifts in organism performance and species interactions with temperature, but it is not clear whether the assumptions of temperature effects on parameters used in the theoretical literature align well with the dynamics observed in the empirical literature. This uncertainty is in part due to potential differences between long- and short-term responses to temperature. To resolve this problem, we could measure species interactions in situ or infer them from the dynamics themselves by determining what parameter values and model structures can reproduce the observed dynamics.

Here we examine the predator–prey cycles of the ciliate *Didinium nasutum* (hereafter *Didinium*) foraging on the ciliate *Paramecium bursaria* (hereafter *Paramecium*) as they pass through a population cycle. The *Didinium-Paramecium* predator prey system is a classic laboratory tool for studying predator–prey dynamics (*Gause, 1934*; *Luckinbill, 1973*; *Salt, 1974*; *Jost & Ellner, 2000*; *Minter et al., 2011*; *Montagnes et al., 2012*; *DeLong & Vasseur, 2013*; *Li et al., 2013*; *Li & Montagnes, 2015*; *Salt et al., 2017*). The advantages of this system are clear: short generation times, small spatial footprint, specialization of *Didinium* on *Paramecium* making for a very strong predator–prey interaction, and ease of estimating population abundances. This system has been used to reveal numerous aspects of predator–prey interactions, and, as with many other ciliates, the *Didinium-Paramecium* system is a useful model for understanding thermal biology across a wide range of temperatures (*Weisse et al., 2002*; *Yang et al., 2013*). The breadth of temperatures that a predator can tolerate is constrained by the breadth of temperature their prey can tolerate, unless alternative prey are available. Therefore, we measured the population dynamics of this system at six temperatures that span almost the full range of positive growth of *P. bursaria* (*Luhring & DeLong, 2017*) to determine how the predator–prey cycle changes with temperature. We then used ordinary differential equation (ODE) fitting to estimate the model parameters (i.e., the mechanisms of change in this system) and show how the net effect of temperature on dynamics is linked to changes in model parameters. Finally, we evaluated how each parameter influences the dynamics on their own. Together, these results indicate a complex suite of temperature effects on both predator and prey that lead to striking variation in ecological dynamics.

## MATERIALS AND METHODS

We acquired *Didinium* from Carolina Biological Supply (Burlington, NC, USA), and we isolated *Paramecium* from a pond at the Spring Creek Prairie Audubon Center southwest of Lincoln, Nebraska, USA (*Novich et al., 2014*). Stock cultures of both species were maintained in the laboratory at 23 °C in medium made from protozoan concentrate (Carolina Biological Supply, Burlington, NC, USA) mixed with filtered and autoclaved pond water acquired from the source pond for *Paramecium* (1:9 ratio of concentrate to water).

We assembled six mL microcosms in 60 mm diameter plastic Petri dishes with lids, with predator and no-predator treatments. We randomly assigned microcosms to predator dishes (six replicates) and predator free dishes (four replicates). For predator dishes, we added 5.9 mL of 40 μm filtered *Paramecium* stock culture (initial density of ~30 cells per mL) to each dish. We rinsed didinia in sterile medium and then added two individuals to
each predator dish in a 0.1 mL aliquot. We added this same volume of rinse medium to each predator-free dish to control for possible microbial contributions from the *Didinium* stock culture to the experimental dishes. Predator free dishes contained 5.9 mL culture medium plus six paramecia transferred in a 0.05 mL aliquot. We assigned replicate dishes randomly to one of six temperatures (17, 20, 23, 25, 27 and 31 °C) and kept them in Percival incubators on a 12:12 h light:dark schedule.

We sampled cultures daily for the first 5 days and then sampled them every 1–3 days until day 18. We took a 0.1 mL sample from each dish daily and replaced this with 0.1 mL sterile medium plus 0.1 mL autoclaved pond water to account for evaporation. We conducted a complete visual census of the *Didinium* population by scanning the entire dish through the microscope. For *Paramecium*, we used a scaled sampling regime, counting the paramecia in the 0.1 mL sample when abundant and conducting a complete census of paramecia in the dishes when they were rarer (~<100 cells) (*DeLong & Vasseur, 2012*). We averaged densities across replicates to create an average trajectory for each temperature. This provides a useful smoothing effect that is often essential for differential equation fitting (*Jost & Ellner, 2000*) and that provides a data set with reduced sampling error and stochasticity induced noise. In a few dishes, the *Didinium* population went extinct within a day or two, reducing our replicate population numbers to 4, 6, 4, 5, 6 and 3 at the temperatures of 17, 20, 23, 25, 27 and 31 °C, respectively. In one replicate at 23 °C, the *Didinium* population increase showed a pronounced lag, while the *Paramecium* population showed a growth, crash, and regrowth, generating deviations in dynamics well beyond the other replicates. We therefore excluded this replicate from the analysis.

At the peak of each *Didinium* population trajectory, we photographed 7–19 individual didinia with a Leica M165C microscope and digital camera, measured cell length and width, and calculated cell volume using the formula for a prolate spheroid.

We used the following ODE model to describe the time series of *Didinium* population density (*C*, for consumer) and *Paramecium* population density (*R*, for resource):

$$\frac{dR}{dt} = (r - r_{\text{slope}}R)R - \frac{aRC}{1 + ahR + m(C - 0.167)} \tag{1a}$$

$$\frac{dC}{dt} = e\frac{aRC}{1 + ahR + m(C - 0.167)} - C(de^{-RC_d}) \tag{1b}$$

In this model, $r$ is the maximum population growth rate of *Paramecium*, and $r_{\text{slope}}$ is the slope of the relationship between population density and realized per capita growth rate. This expression is equivalent to the logistic growth model, where $r_{\text{slope}} = r/K$, with $K$ being carrying capacity, and is the typical form of population growth for *Paramecium* (*Jiang & Kulczycki, 2004*; *Gibert et al., 2017*). We chose this expression because fitting routines converge more easily with it than with the logistic growth model. The two equations are linked by a type II functional response (*Beddington, 1975*; *DeAngelis, Goldstein & O'Neill, 1975*; *Skalski & Gilliam, 2001*), where $a$ is the space clearance rate of the *Didinium* (i.e., the volume of habitat cleared of prey per predator per time), $h$ is the handling time for *Didinium*, $m$ is interference competition among *Didinium*, $e$ is the

efficiency of converting *Paramecium* into new *Didinium*, *d* is the *Didinium* maximum death rate, and $C_d$ sets the density-dependence of death rate. The C-0.167 term allows interference to go to zero when there is only one predator (the density is 0.167 individuals per mL when there is one predator in the dish). Several previous works suggest the necessity of including prey-dependent mortality for *Didinium* dynamics (*Minter et al., 2011*; *DeLong, Hanley & Vasseur, 2014*; *Li & Montagnes, 2015*), and the function introduced here allows mortality rates to decline as prey become more abundant.

Despite the fact that we cannot know with certainty what the right model is for the *Didinium-Paramecium* interaction, our chosen model contains key components of all consumer–resource interactions and displayed a high degree of compatibility with the data (see Results), suggesting the model reflects real aspects of the thermal biology of this system. In general, ODE models are particularly useful for protist microcosm dynamics, since cell division and death can happen at any time and reproduction does not occur during discrete breeding periods. However, ODE models have the downside of often predicting exceedingly low population abundances (much less than one individual) from which populations can still rebound. These low abundances are generally taken to reflect densities for populations with large spatial scales. In the case of whole populations contained within microcosms, these low abundances can be thought of as functionally zero.

We fit Eq. (1) to the time series data using the Potterswheel toolbox version 4.1.6 in MATLAB 2017a (*Raue et al., 2009*; *DeLong, Hanley & Vasseur, 2014*). We used the average time series rather than individual replicate populations to avoid fitting stochasticity and noise in the data sets and to aid in the identification of confidence intervals on the parameters. Fits to individual replicate time series were possible for some replicates but not others, as stochasticity and limited number of observations made identifying a good model and robust parameter estimates impossible in some cases. Furthermore, differences among replicates arose through stochasticity, such that the variation across replicates reflects variation not caused by the underlying deterministic drivers of the system, and it is these underlying mechanisms that are of interest in this study. The Potterswheel fitting tool searches parameter space to identify parameter sets for which the solution to Eq. (1) provides a fit to the data for which further changes in parameters do not lead to an improved fit. The fitting approach minimizes a $\chi^2$ deviance across all measurements.

We used profile likelihood estimation to characterize uncertainty of the parameter estimates (*Raue et al., 2009*). Profile likelihood calculates the parameter value for which an increase in the model's $\chi^2$ goodness of fit statistic reaches a particular threshold. We set this threshold to a 68.5% confidence interval (CI) because pushing the parameters farther than this from the mean estimate frequently caused integration failure of the solvers. Thus, our uncertainty estimates are approximately the standard deviation of the parameter (*Raue et al., 2009*). The profile-likelihood is estimated in log increments, preventing negative confidence interval estimates. We prioritized profile likelihood confidence intervals, but in cases where these were unattainable, we used CIs estimated with the Hessian matrix of Eq. (1) provided by the Potterswheel toolbox.

We first fit Eq. (1) to each time series with all parameters unconstrained. This process indicated that the conversion efficiency (parameter $e$) was very similar across temperatures (mean = 0.055, with CIs 0.028–0.097 inclusive of all temperatures). We tested whether fixing the conversion efficiency to the mean value impaired fits, and at all temperatures, $\chi^2$ values increased by only 0.2–6.4, indicating little impact on fit quality. We therefore inferred that conversion efficiency is somewhat invariant with temperature. We also determined that the density-dependent mortality parameter ($C_d$), while necessary to include given poor fits without it, was nonetheless very difficult to estimate. We detected good fits in the area of $C_d = 40$, and we also determined that $C_d$ could be fixed at 40 without loss of fit quality.

Previous evaluations demonstrated that the Potterswheel ODE fitting approach provides robust parameter estimation without generation of spurious covariation among parameter estimates (*DeLong et al., 2018*). Nonetheless, we evaluated the ability of our ODE fitting methods to recover model parameters from dynamics. We did this in two steps. First, we used ODE solvers to generate simulated model dynamics from Eq. (1), and then second, we used the same fitting routines as used in the main analysis to identify model parameters of these simulated datasets. We repeated the testing for each of the six estimated parameter sets (one set for each temperature). We also restricted the timespan of the test simulation to the time frame of the observed dynamics, which includes all time steps with positive abundances and the three time periods of zero abundance after the last non-zero abundance time point. At all six temperatures, the ODE fitting returned exactly the parameters used to generate the simulations (Fig. S1). We repeated this analysis after introducing noise to the time series by adding a number drawn from a random normal distribution ($\bar{x} = 0$; $\sigma = 0.1$) and repeating the fitting process at each temperature 10 times. We were still able to recover parameters, albeit less exactly than without noise (Fig. S2). All MATLAB files required to conduct fitting and test parameter recovery are available in the Supplemental Materials.

*Paramecium* populations in the predator free control dishes increased in density through time, but these populations did not achieve an identifiable carrying capacity at all temperatures. We therefore used this data only to calculate rate of growth $r$ for *Paramecium* in predator free conditions. We extracted data from days 2 and 4 and used the standard exponential growth model: $r = \frac{\ln \frac{N_4}{N_2}}{2}$, where $N_2$ and $N_4$ are population densities at time 2 and 4, respectively. There was a decline in density from the initial inoculation to day 2, so we did not use this first time step. We calculated $r$ for each replicate separately to estimate error.

Finally, we evaluated the effects of variation in parameters due to temperature on the dynamics. We first solved our model for the mean parameter set. We then varied each parameter on its own from the minimum to the maximum fitted values across temperatures and solved the model again using the mean fitted values for the other parameters. Thus, for each parameter we show three sets of dynamics reflecting three parameters sets: (1) the minimum parameter set contains the minimum fitted value of the focal parameter and the means for the other parameters, (2) the mean parameter set uses

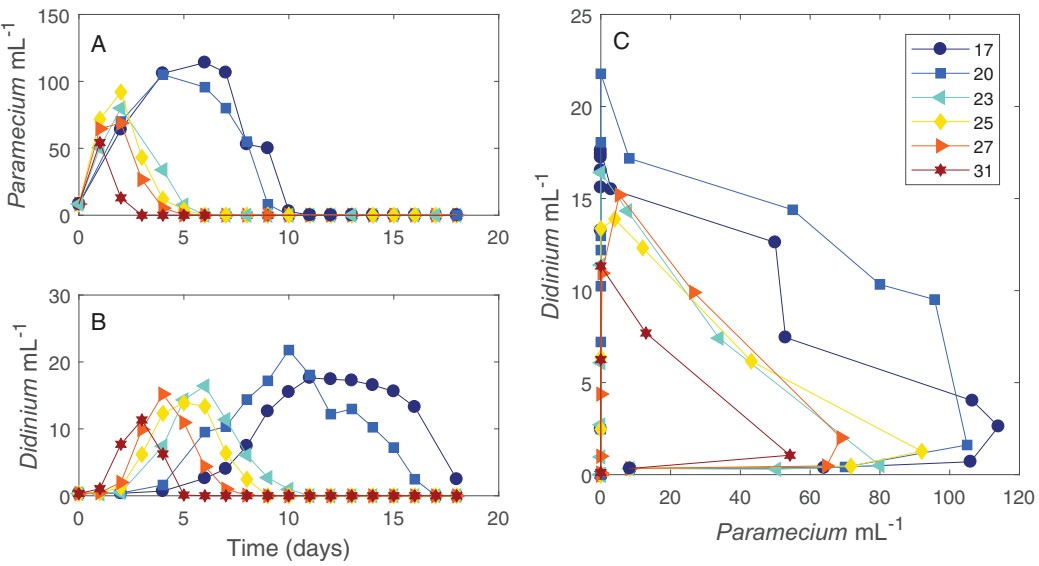

**Figure 1 Dynamics of interacting *Didinium nasutum* and *Paramecium bursaria* populations.** Points are population densities averaged across replicates on each day. (A) and (B) show how the dynamics change for each population with temperature separately, with (A) showing *Paramecium* and (B) showing *Didinium*. (C) shows the state space trajectories for the populations together.

the mean of all parameters and thus is the same in all contrasts, and (3) the maximum parameter set contains the maximum fitted value of the focal parameter along with the means of all other parameters. These contrasts show how variation in specific parameters alters the predator–prey dynamics while holding the other parameters constant.

## RESULTS

The dynamics of the interacting *Didinium-Paramecium* populations showed a clear shift in shape from colder to warmer temperatures (Figs. 1 and 2). As temperature increased, the period of the population cycle decreased, while the amplitude of the *Didinium* cycle increased and then decreased and the amplitude of the *Paramecium* cycle started to decline around 23 °C (Figs. 1A and 1B). In state space, these shifts were seen in a reduction in the radius of the trajectory as populations increased, decreased, and finally went extinct (Fig. 1C). The time to extinction decreased as temperature increased (Fig. 1).

Based on our fitting results, we infer that six of the parameters governing these interactions changed with temperature (Fig. 3). Most parameters (*Paramecium* growth rate, *Paramecium* strength of density dependance, space clearance rate, interference, and handling time) showed a unimodal response, peaking at intermediate temperatures, typically near 27 °C, but at 20 °C in the case of handling time. In contrast, maximum *Didinium* mortality showed a monotonic increase with temperature. Finally, in the predator free dishes, *Paramecium* rate of population growth *r* peaked at 23 °C, but overall, rate of growth for *Paramecium* was much lower in the predator-free dishes than in the presence of predators, especially at higher temperatures (Fig. 3A).

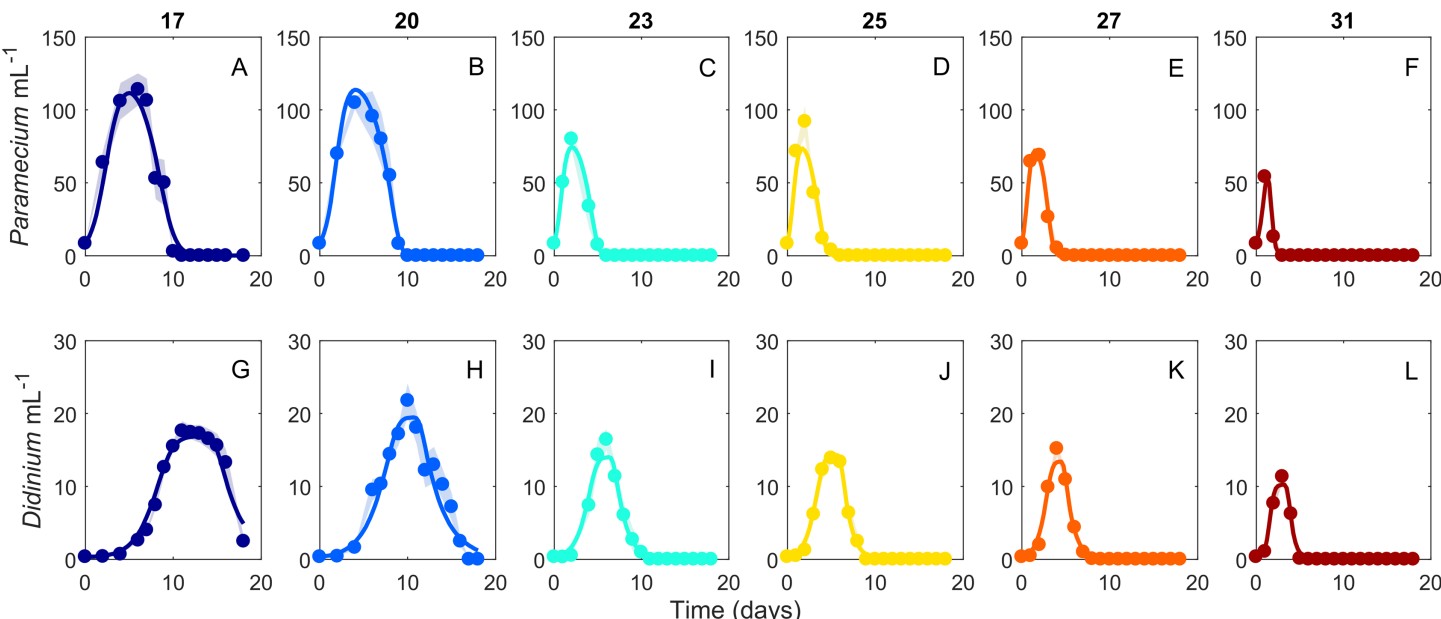

**Figure 2 Dynamics of interacting *Didinium nasutum* and *Paramecium bursaria* populations.** Points are population densities averaged across replicates on each day. (A–F) are *Paramecium bursaria* and (G–L) are *Didinium nasutum*. Temperatures are shown from cool to warm colors from left to right. Shaded areas represent SE of the across-replicate mean at each time point, and the heavy lines show fits of our model (Eq. 1) to the data.                                                                                               

The shifts in dynamics in this system due to changes in temperature were influenced strongly by all temperature-dependent parameters (Fig. 4). The changing amplitude in the system (Fig. 1) was influenced by variation in *Paramecium* rate of growth and density dependance as well as *Didinium* mortality. The shifts in period were more related to a combination of the effect of temperature on space clearance rate, mutual interference, and handling time (Figs 4G and 4H).

*Didinium* cell volume shifted with temperature. From the initial size ($4.8 \times 10^{-4}$ $\mu m^3$ in stock cultures maintained at 23 °C), mean *Didinium* cell volume increased with temperature from 17 to 21 °C and then decreased with further increases in temperature (Fig. 5).

## DISCUSSION

Predicting how changes in temperature alter population and community dynamics depends on developing a thorough understanding of how temperature alters the underlying drivers of population growth and species interactions. Currently, however, numerous assumptions about the temperature dependance of parameters governing species interactions are still required to make predictions about the effects of warming on population and community dynamics. A more complete depiction of the effect of temperature on underlying mechanisms driving population dynamics is needed, especially over broad temperature ranges and taking into account organismal acclimation. Here we used a combined theoretical-empirical approach to characterize dynamics across temperature and uncover the temperature dependance of the drivers of these patterns.

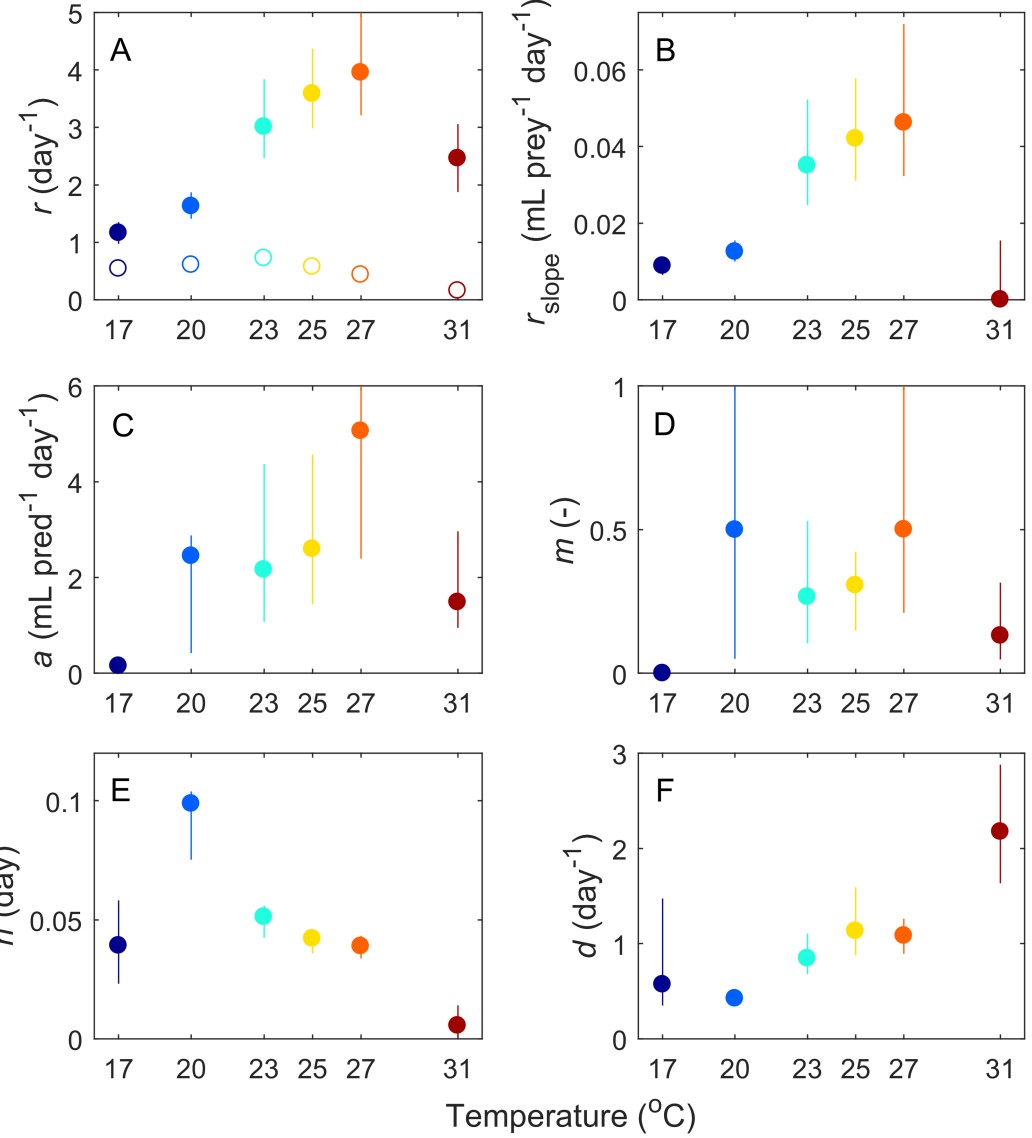

**Figure 3 Estimated parameters from the fits of our ODE model (Eq. 1) to the average dynamics of *Didinium nasutum* consuming *Paramecium bursaria*.** Solid dots indicate estimate and vertical lines indicate 68.5% confidence intervals (equivalent to one standard deviation). Parameters are (A) intrinsic rate of growth of *Paramecium*, (B) strength of density dependance in *Paramecium*, (C) space clearance rate, (D) interference, (E) handling time, and (F) conversion efficiency. In (A), open circles are the measured intrinsic rate of growth for *Paramecium* in the predator-free dishes.

We found that the dynamics of the *Didinium-Paramecium* system shifted steadily as temperature increased, from longer to shorter cycle periods and a change in amplitude (Figs. 1A and 1B). This outcome is somewhat similar to the increase in cycle amplitude shown by tea tortrix moths (*Adoxophyes honmai*) as a result of increasing temperature (*Nelson, Bjørnstad & Yamanaka, 2013*), to the decreasing period and increasing amplitude of pea aphid (*Acyrthosiphon pisum*)-parasitoid wasp (*Aphidius ervi*) cycles with increasing temperature (*Meisner, Harmon & Ives, 2014*), and to shifts in both

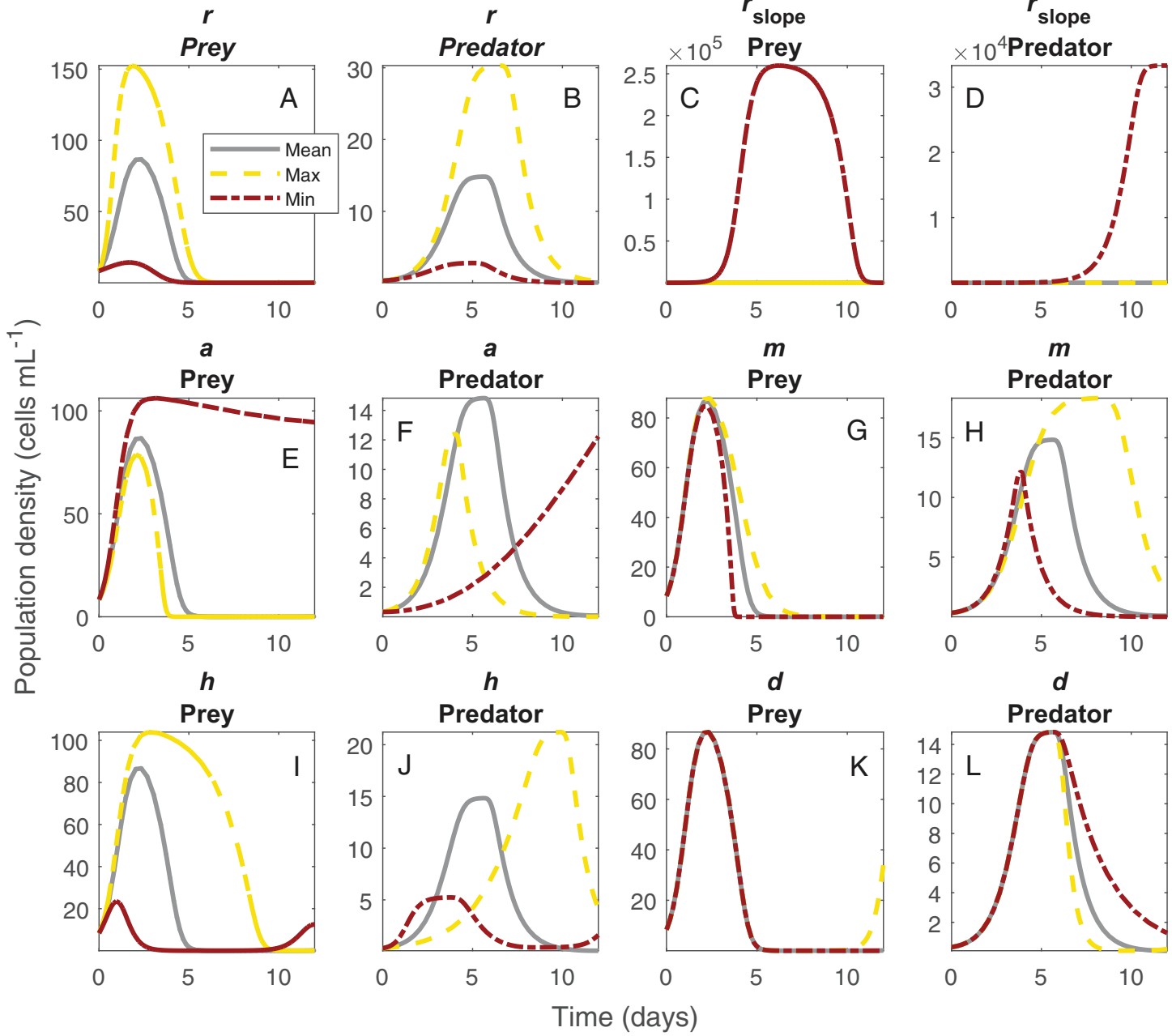

**Figure 4 Changes in population dynamics due to variation in model parameters.** Each set of panels shows variation in population dynamics caused by variation in a single parameter, with other parameters held constant at the average of the fitted values across temperature. Panels show intrinsic rate of growth of *Paramecium* for prey (A) and predators (B), strength of density dependance in *Paramecium* for prey (C) and predators (D), space clearance rate for prey (E) and predators (F), mutual interference for prey (G) and predators (H), handling time for prey (I) and predators (J), and *Didinium* maximum mortality rate for prey (K) and predators (L). The lines show the trajectories for the *Paramecium* and the *Didinium* populations for the mean parameter set (gray lines), the maximum parameter set (yellow lines), and the minimum parameter set (brick red lines). The maximum and minimum estimated parameters do not generally correspond to the extremes of temperature.

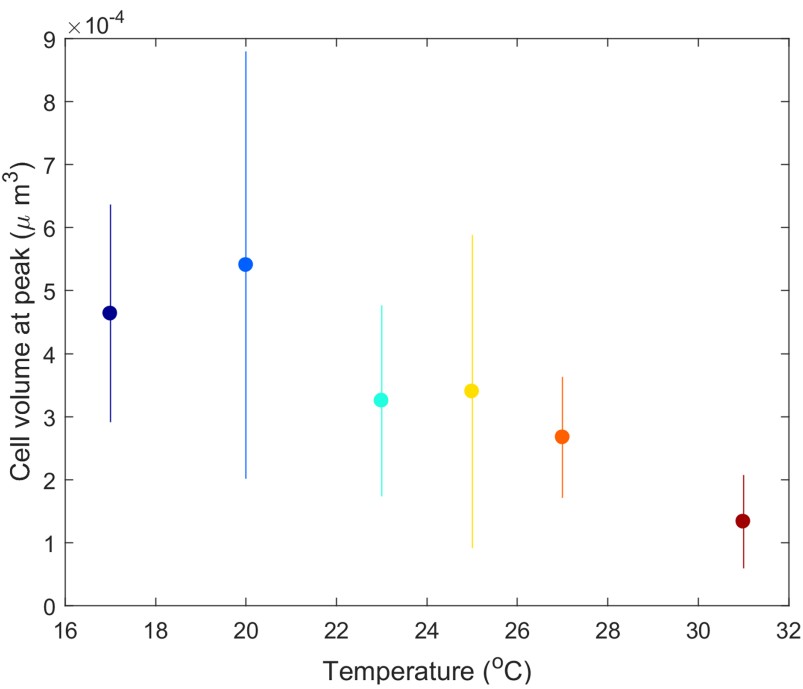

**Figure 5 Mean *Didinium nasutum* cell volume at the top of their population cycle for each temperature.**

period and amplitude of *Didinium-Paramecium caudatum* dynamics (*Salt et al., 2017*). Yet by considering a broad range of temperatures, we also found that some of these dynamic shifts themselves may be unimodal rather than monotonically changing.

Our ODE fitting analysis revealed that the underlying drivers of these dynamics showed pervasive responses to temperature (Figs. 1 and 2). Most components of our model, including prey births, the functional response and predator mortality, showed some type of response to temperature (Fig. 3). In contrast to expectations about the exponential (Arrhenius type) effect of temperature on ecological processes (*Brown et al., 2004*; *Dell, Pawar & Savage, 2011*; *Burnside et al., 2014*), only *Didinium* mortality rate showed an exponential-like increase from low temperature to higher temperatures. Rather, most parameters showed a unimodal response, in line with other observations about fecundity, population growth, and the functional response across temperatures (*Ratkowsky, Olley & Ross, 2005*; *Englund et al., 2011*; *Amarasekare, 2015*; *Uszko et al., 2017*; *Uiterwaal & DeLong, 2020*). The typical peak or minimum temperature was 27 °C, such that four of the five parameters shifted in a correlated manner as temperature changed. This pattern also reflects the previously observed correlation between space clearance rate and interference in the *Didinium-Paramecium* system (*DeLong & Vasseur, 2013*). Thus, the actual dynamics of these populations arose from a combination of monotonically increasing and unimodal responses of ecological functions, which is consistent with some theoretical work (*Amarasekare, 2015*). By varying parameters one at a time, it appears that temperature effects on birth and death rates influenced changes in cycle amplitude, while temperature effects on the functional response influenced changes in cycle period (Fig. 4).

Our approach here was to infer the temperature dependance of the mechanisms driving population dynamics by fitting an ODE consumer-resource model (Eq. 1) to data and comparing fitted parameters across temperature (*DeLong et al., 2018*). Although all parameter estimates are model-dependent and thus to some degree sensitive to model selection, this approach (the indirect approach, sensu *Palamara et al. (2014)*) allowed us to uncover temperature dependencies that unfolded over the full period of the interaction. Although the shape of some of these relationships were quite clearly unimodal (e.g., maximum population growth rate), other cases were less clearly so (e.g., space clearance rate), suggesting the need for additional work to identify these relationships more precisely (Fig. 3).

The predator–prey interaction in our experiment lasted about 3–25 *Didinium* generations (maximum generation time estimated as $e/h$) and 20–80 *Paramecium* generations (generation time estimated as $1/r$). Over these generations, changes in *Didinium* or *Paramecium* physiology or morphology, through acclimation or phenotypic plasticity, could have had time to take effect and influence the outcome. For example, *Didinium* in our study showed a clear dependance of cell volume on temperature at the peak of their cycle (Fig. 5), partially consistent with the temperature-size rule (*Atkinson, 1994*; *DeLong, 2012*), although cell volume likely was also responding to the dynamic changes in resource levels through time (*DeLong, Hanley & Vasseur, 2014*). Thus, our results reflect long-term rather than short-term responses to temperature.

In our study, both predator and prey populations went deterministically extinct by the end of the experiment at all temperatures, consistent with the typical behavior of *Didinium* and *Paramecium* without stabilizing environmental factors introduced to the microcosms (*Luckinbill, 1973*; *DeLong & Vasseur, 2013*; *Salt et al., 2017*). Although extinctions occurred earlier in warmer temperatures, the system did not undergo major qualitative shifts such as from stable to unstable or from having a fixed point equilibrium to having oscillations, which are common predictions from theory (*Vasseur & McCann, 2005*; *Amarasekare, 2015*). Rather, temperature somewhat smoothly compressed the dynamics from a longer cycle through state space to one that lasted only a few days (Fig. 1C).

We selected Eq. (1) as a plausible model to use in our fitting routines. Although other types of models could work, we emphasize that most of the terms and parameters in our model are fundamental to predator–prey interactions and must be included in any type of model fitting in some manner. Parameterizing models such as Eq. (1) with data taken from single-species laboratory observations is a useful way of dealing with the challenge of complex fitting problems such as the one here (*DeLong, Hanley & Vasseur, 2014*). We conducted predator-free growth experiments for paramecia to identify some parameters outside of the ODE fitting task, but it was clear that in the presence of the predator, paramecia underwent cell divisions at a much higher rate early in the experiment, especially at higher temperatures (Fig. 1A). Thus, we could not fix *Paramecium* rate of growth with data from the single-species cultures to make fitting easier and improve our estimates of other parameters. This difference is consistent with recent observations that population growth thermal performance curves shift in the presence of predators (*Luhring & DeLong, 2016*), but it suggests that predicting the effects of

temperature on predator–prey dynamics without in situ parameter estimates could prove misleading.

The difference between control and predator treatment *Paramecium* maximum rate of growth ($r$) also was unexpected since the implied division rate of *Paramecium* at low densities was unusually high for a *Paramecium*-sized protist. We would anticipate that such a high $r$ would strictly occur at the lowest densities, with realized growth rates being much lower at even slightly higher population densities. It is also possible that the high growth rate was linked to short-term bursts of cell divisions arising from stored nutrients carried over from stock conditions or from reductions in cell volume. If this latter possibility occurred, smaller cells could alter the conversion efficiency of the predators, leading to compensating effects on the estimates of $r$. Untangling such shifts in parameters through time, and their impact on fitted parameter estimates, may require greater effort to quantify traits such as cell size and processes such as foraging rates in situ while tracking densities.

## CONCLUSIONS

In conclusion, our results imply that there are few parameters that can be overlooked when seeking to predict shifts in population dynamics with changes in temperature. Although it is clear that shifts in the variance of temperature are likely to influence population outcomes (*Estay, Lima & Bozinovic, 2014*; *Vasseur et al., 2014*), it is also clear that shifts in mean can have profound effects on community dynamics through direct effects on all of the mechanisms underlying such patterns. Our analysis focused on a specialized predator foraging on a single type of prey, but such tight species interactions might be quite rare in nature, with many predators having considerably broader diets than *Didinium*, including some protists (*Roberts et al., 2010*). It is therefore still unknown how the patterns of temperature dependance we observed here would manifest in a more diverse community with more generalist consumers. Nonetheless, as evidence amasses that changes in climate are altering the structure of the natural world and the dynamics of populations (*Parmesan & Yohe, 2003*; *Root et al., 2003*; *Tylianakis et al., 2008*; *Van De et al., 2010*; *Beck-Johnson et al., 2013*), an assessment of the full range of temperature dependent mechanisms is needed.

### Funding
This work was supported by a James S. McDonnell Foundation Studying Complex Systems Scholar Award. The funders had no role in study design, data collection and analysis, decision to publish, or preparation of the manuscript.

### Grant Disclosures
The following grant information was disclosed by the authors:
James S. McDonnell Foundation Studying Complex Systems Scholar Award.

## Competing Interests

The authors declare that they have no competing interests.

## Author Contributions

- John P DeLong conceived and designed the experiments, performed the experiments, analyzed the data, prepared figures and/or tables, authored or reviewed drafts of the paper, and approved the final draft.
- Shelby Lyon conceived and designed the experiments, performed the experiments, authored or reviewed drafts of the paper, and approved the final draft.

## Data Availability

The MATLAB script for reading, analyzing, and plotting data, Potterswheel functions for ODE fitting, and the data on the abundances of *Didinium* and *Paramecium* and Didinium cell sizes and fitted model parameters are available in the Supplemental Files.

## Supplemental Information

Supplemental information for this article can be found online at http://dx.doi.org/10.7717/peerj.9377#supplemental-information.

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
