# Peer review of "Temperature alters the shape of predator–prey cycles through effects on underlying mechanisms"

_PeerJ, doi:10.7717/peerj.9377_

## Round 0.1 · original submission · Major Revisions

Many thanks for submitting this study, which was a pleasure for me to read. The two reviewers, while positive about the research questions and the general approach, outline a number of limitations and issues that indicate that the manuscript cannot be accepted in its current state. They have provided constructive comments on how the issues could be solved. If a revised manuscript is submitted, it will go out for review again.

·

Basic reporting

see below

Experimental design

see below

Validity of the findings

see below

Additional comments

General:
This study addresses important issues in a useful and original way. I am fully supportive of the questions asked, the approach used to answer them, and (mostly) the findings and their interpretation. I start with my main (and really only) concern. I then provide, following the Journal’s guidelines, some other comments, for which I have hopefully indicated possible changes.
Main concern

Paramecium is unlikely to ever be able to divide more than two time in a day (i.e. maximum growth rate is 1.4 d-1). It’s just not built to do more. Some ciliates, when faced with a nutritional shift down will undergo rapid divisions without an increase in size (e.g. see Lynn et al. 1987, Divider size and the cell cycle after prolonged starvation of Tetrahymena corlissi. Microb Ecol 13:115-127). So, then let’s say Paramecium might have divided 3 times in a day, and growth rate would be 2 d-1, under transient conditions. You, however, report apparent growth rates of up to 5 d-1 from your ODE (Fig. 3 a), and these differ substantially from the independently determined estimates (without predators) and from those presented in the literature (e.g. Salt et al 2017,which you cite, but also see Krenek et al. 2011, Thermal performance curves of Paramecium caudatum: a model selection approach. European Journal of Protistology. 2011 47:124-37.)

Where does that leave us? I’m no expert on the method you used to obtain parameters from ODE, but it strikes me that if one of your parameters is way off, then others will be affected. Is this so, and if so, how will it alter your other parameter estimates and conclusions? Would it not be possible to constrain some of the parameter estimates (not just prey growth rate) using literature values and your own experimental values? Would this not then provide more realistic estimates and trends?

Have I missed something here? If I'm right, then the study might require a major overhaul,and if I'm wrong, then I only ask the Authors to:
1. explain why it's ok to have these high growth rates (I don't really accept the current explanation in the paper that Paramecium was growing at these rates)
2. think about the other changes/additions I've suggested.

Clear and unambiguous, professional English used throughout:
Generally, the manuscript was well written and clear. There were a few very minor issues:
Line 118: is this “Paramecium” or “Paramecia”?
Line 132: missing a degree symbol before C.
Line 140: you might indicate why C and R are used (presumably Consumer and Resource?
Line 150: I don’t think this is actually “volume cleared of prey…”. It is volume processed by a consumer per unit time x the efficiency of capturing a particular prey type (i.e. it is prey dependent). Few people note this distinction, but it IS worth doing so, I feel.
Line 161: change “challenging” to “impossible”? If it was only challenging, why did you not rise to it?
Line 162: “arose” not “arise”?
Line 172: delete “that”
Line 192 (and elsewhere): delete “in order”: “to” is sufficient.
Line 208: did it really “increase”; the data are not so strong here, I feel.

General: I also felt that the Discussion could be tightened up:
Lines 307-312; could this information be moved to the Methods?
Lines 324-331; I feel this could have been reduced and put into the Introduction by saying what this model can do, rather than ending with what it cannot do. However, these are stylistic issues, that the Authors can ignore if they wish. I just feel that these changes might make it a punchier paper.

Finally: I felt that in a few places in the Discussion, the obvious was stated. These related to a unimodal or monotonic increase, rather than an exponential increase. If the underlying experimental design is based on the full consumer thermal response (which is almost by definition unimodal (and right-skewed), then I think we might expect most underlying responses to be unimodal. I think this could be explored in a more rigorous fashion, rather than just stating it (see literature below for some thoughts)

Literature references, sufficient field background/context provided:
I think that the use of the literature is sufficient, but I would like to give the Authors the opportunity to consider the studies below, as they may provide additional insights and useful background on temperature-food concentration interactions, model elaborations, and predator-prey dynamics of protists. Clearly, the Authors need not cite any of these, and they may be aware of all of them, but I know I miss stuff, so here it is:

Montagnes DJS, Zhu X, Gu L, Sun Y, Wang J, Horner R, Yang Z (2019) False exclusion: a case to embed predator performance in classical population models. American Naturalist. 194:654-670
Wang Q, Lyu Z, Omar S, Cornell S, Yang Z, Montagnes DJ. (2018) Predicting temperature impacts on aquatic productivity: Questioning the metabolic theory of ecology's “canonical” activation energies. Limnology and Oceanography. 64: 1172-1185doi.org/10.1002/lno.11105
Yang Z, Zhang L, Zhu X, Wang JK, Montagnes DJS (2015) An evidence-based framework for predicting the impact of differing autotrophic-heterotroph thermal sensitivities on consumer-prey dynamics, ISME Journal 10:1767–1778 doi: 10.1038/ismej.2015.225
Yang Z, Lowe CD, Crowther W, Fenton A, Watts PC Montagnes DJS (2013) Strain-specific functional and numerical responses are required to evaluate impacts on predator-prey dynamics ISME Journal 7:405-416
Cooper JK, Li J, Montagnes DJS (2012) Intermediate fragmentation per se provides stable predator-prey metapopulation dynamics. Ecology Letters 15:856-863
Montagnes DJS, Roberts E, Lukeš, J, Lowe C (2012) The rise of model protozoa. Trends Microbiol. 20:184-91
Kimmance S, Atkinson D, Montagnes DJS (2006) Do temperature - food interactions matter? Responses of production and its components in the model heterotrophic flagellate Oxyrrhis marina. Aquat Microb Ecol 42:63-73 66.
Atkinson D, Ciotti BJ, Montagnes DJS (2003) Protists decrease in size linear with temperature: ca. 2.5 ºC-1. Proc Royal Soc 270:2605-2611
Montagnes DJS, Kimmance SA, Atkinson D (2003) Using Q10: can growth rates increase linearly with temperature? Aquat Micob Ecol 32:307-313 71.
Weisse T, Stadler P, Lindström ES, Kimmance SA, Montagnes DJS (2002) Interactive effect of temperature and food concentration on growth rate: a test case using the small freshwater ciliate Urotricha farcta. Limnol Oceanogr 47:1447-1455 78.
Montagnes DJS, Weisse T (2000) Fluctuating temperatures affect growth and production rates of planktonic ciliates. Aquat Microb Ecol 21:97-102 83.
Weisse T, Montagnes DJS (1998) Effect of temperature on inter- and intraspecific isolates on Urotrichia (Prostomatida, Ciliophora). Aquat Microb Ecol 15:285-291

Professional article structure, figures, tables:
I wonder if Figs 1 and 2 could be combined? Just a thought.

Reviewer 2 ·

Basic reporting

No comment.

Experimental design

No comment.

Validity of the findings

No comment.

Additional comments

It was my pleasure to read and review the article “Temperature alters the shape of predator-prey cycles through effects on all of underlying driving mechanisms”. It deals with a very timely subject – how temperature affects ecological dynamics through changes in underlying mechanisms that govern populations of interacting prey and predator. The authors used a novel method of direct parameter fitting to an ordinary differential equation model, a method that is very powerful and allows for a reliable characterization of mechanistic system behavior.

I do have however a number of comments that mostly focus on model formulation, model choice, and subsequently on parameter interpretation. Many of this issues are somewhat connected, but still I tried to list them one-by-one below.

1. My first concern is the model formulation. I think that it is rather complicated for the given system with its simple dynamical pattern (i.e. one abundance peak followed by extinction). An overly complex model could create problems with parameter redundancy, leading to a situation in which our confidence in fitted parameter values is low as many of them are strongly statistically correlated. The authors partly took care of that issue by performing a simple model selection procedure. However, this in turn created another issue – that estimates of each parameter, when compared across temperature gradient, are compared between different models, and each model formulation changes (in a statistical sense) the interpretation of parameters as they correlate.

First, using the parameter r_slope (=r/K) instead of just the carrying capacity K produces a very strong correlation between parameters r and K. Such situation should be avoided as this hinders us in identifying a single set of mechanistically interpretable parameter estimates. This could be, for instance, the reason why r and K behave very similarly across temperature (Fig. 3 A and B, full circles) – I believe they do that because there is no other choice when strongly correlated. Also, if r_slope is close or equal to 0, it is hard to know what it should mean for parameter r. Most likely, this means that there is not enough data in the time series in which the prey gets sufficiently abundant so that K can be identified. This could also be the real reason why estimates of r differ between predator-present and predator-absent treatments – when predator is present, the prey never gets very abundant therefore the model fitting “does not see” the carrying capacity, therefore the estimate of r gets high (Fig. 3 A).

Second, the formulation of prey-density-dependent predator mortality is somewhat awkward. Note that, in principle, such formulation means that there is an extra prey-density-dependent growth term for the predator “+(d*R)/(Cd+R)”. It feels not needed, as predator growth is already, through the functional response, a positive function of prey density. Then, if there is any mortality in the predator during experiments, it means that parameter d has to be positive. This in turn would force parameter Cd to be non-zero and identifiable as well, because otherwise the whole mortality term would cancel out. And this latter option is actually what seems to be happening here – the estimates of Cd are equal to 0 or very close to it (Fig. 3 H) so that the mortality cancels out – and this means there was virtually no Didinium mortality during the experiments! This might mean that the estimates of d are just statistical artifacts as anyway the whole term “-d+(d*R)/(Cd+R)” is equal to 0.

As the best way to deal with these issues, I would recommend to run a full model selection. That is, compare all possible model combinations, with the current Full Model on one end, and a simple Lotka-Voltera model (with type I functional response, no predator interference, and no predator mortality) on another end. In addition, it is important to select a model that gives the lowest AICc values at all temperatures combined. This will ensure that parameter estimates are compared within the same model across all temperatures, therefore within a single interpretation of the parameters. Only then your statement in the discussion section (lines 308-312) can be considered true, that is, the chosen model would truly contain only the essential parameters.

2. Another thing that got me worried is the fact that the authors used replicate averaging, and that they added extra interpolated time series points. I understand that this might be needed in order for the fitting procedure to work, however in principle such things should not be done. I wonder if this issue would remain if a simpler model is fitted. Then, as parameters would be more strictly defined and hopefully less correlated, averaging and extra data points might not be needed anymore. It is also possible that high uncertainty is an immanent property of protist predator-prey systems, and in that sense it is important to show it to the readers.
Either way, it can also mean that the authors are reaching the limit of what the ODE fitting method can achieve. Perhaps with that many time series data points, and for that many parameters to be estimated, it just does not work satisfactorily well. It might be a good idea to highlight this potential limitation in the discussion section.

3. I am slightly sceptic to the story about long-term warming effects. Obviously, acclimation, adaptation, selection etc. can be happening in the studied experimental system. However, with the current modeling approach, all such things are treated as implicit processes inside the black box of population-level mechanisms, brought together by a small bunch of fixed, non-evolving parameters. In other words, model parameters are explicit functions of temperature only, so that other effects (long-term or not) can be only seen as the net effect. The authors mention that in the discussion, however when I read the abstract and the introduction, I almost expected to see an eco-evo model. It might be a good idea to tone down a bit this story early in the manuscript.

4. More of semantics and interpretation issue: If the predator-prey cycles shrink in state space with warming, it should mean that the cycle amplitude also has to shrink. In Fig. 1 B it seems like the Didinium cycle first goes up and then down with its amplitude, but it is not very clear, and Paramecium does not do that. I would probably not mention the unimodal response of the amplitude to warming, but just stick to the observation that it generally declines, and so the state space of the cycle shrinks. Additionally, as there is only one cycle peak, I am not sure how informative it is to talk about the cycle period defined as the distance between peaks.

5. What did Paramecium feed on during the experiments? Could the dynamics of this food item be in any way temperature-dependent?

6. The first paragraph in the discussion section (lines 235-249) can probably be completely removed from here, with some parts incorporated to the introduction instead.

---

## Round 0.2 · Major Revisions

Many thanks for the revised manuscript. It was sent out for review with the original reviewers. Due to some remaining concerns, we decided to also send it out to two further reviewers with expertise in ODEs and model fitting. You will find attached all four reviews. All are positive but contain very concrete recommendations. In particular, reviewer 4 recommends an additional simulation with the purpose of validating the modelling approach, which I feel would be the most effective way of dispelling concerns raised here and in previous review rounds.

·

Basic reporting

The Authors have fully addressed all my concerns except the first one, which I still would like them to consider a bit further in the manuscript.

If possible, I would like them to add the following to the Discussion:

1. A clear recognition that for short periods, their estimates of Paramecium maximum growth rate are about 4 times higher than has ever been reported and much higher than would be expected for ciliates of this size.
2. A range of explanations for these that could include:
i. Incomplete parameterisation of the model (i.e. something was missing such as shifts in cell size of either Paramecium or Didinium, which would affect conversion efficiency).
ii. That the high grow rate was real and could have been due to the presence of predators (a very novel and interesting speculation).
iii. The growth rates may have been short-term (transient) effects due to a burst of synchronized divisions.
iv. Anything else the Authors can think of.
3. It might also be useful for the Authors to include a caveat about how estimates are model-dependent, and the cross-temperature pattern is the important part.

Experimental design

see abovesee

Validity of the findings

see above

Additional comments

The Authors have fully addressed all my concerns except the first one, which I still would like them to consider a bit further in the manuscript.

If possible, I would like them to add the following to the Discussion:

1. A clear recognition that for short periods, their estimates of Paramecium maximum growth rate are about 4 times higher than has ever been reported and much higher than would be expected for ciliates of this size.
2. A range of explanations for these that could include:
i. Incomplete parameterisation of the model (i.e. something was missing such as shifts in cell size of either Paramecium or Didinium, which would affect conversion efficiency).
ii. That the high grow rate was real and could have been due to the presence of predators (a very novel and interesting speculation).
iii. The growth rates may have been short-term (transient) effects due to a burst of synchronized divisions.
iv. Anything else the Authors can think of.
3. It might also be useful for the Authors to include a caveat about how estimates are model-dependent, and the cross-temperature pattern is the important part.

Reviewer 2 ·

Basic reporting

No comment.

Experimental design

No comment.

Validity of the findings

No comment.

Reviewer 3 ·

Basic reporting

no comment

Experimental design

no comment

Validity of the findings

no comment

Additional comments

## Summary ##

In this paper the authors use combination of microcosm lab experiments and mechanistic ODE modelling to investigate how predator–prey cycles are affected by temperature. They find that both predators and prey cycle periods shorten with increasing temperatures, and that the amplitudes of cycles shifted with temperature as well.

## General ##

The questions around how predator–prey dynamics shift with temperature are interesting, and the approach taken by the authors to combine experiments with dynamical modelling is very refreshing. The authors' findings are well put in the context of the literature, and their investigations are for the most part well explained. Barring some technical details I believe have to be further discussed by the authors, I believe this manuscript would make a good contribution to PeerJ.

Main comment 1: My main concern regarding this manuscript is around some of the assumptions going into the modelling framework, leading to some interrelated potential problems.

1a) The set of ODEs describing the predator–prey dynamics can, given positive initial conditions, never reach zero density in finite time, whereas virtually all the experiments conducted by the authors did. Furthermore, ODE models are usually approriate in situations with sufficiently high densities, but here the initial conditions for the predators are just 2 individuals. While I certainly don't think this sinks the authors' approach, I do think that these approximations should be mentioned and justified.

1b) Another problem relating to necessary positive solutions of the ODEs is the structure of variance. Typically in situations like these, one would assume that the solution can be captured by some deterministic, mechanistic part, plus some random variation around this mean, so that e.g., y(t) = μ(t) + N(0,σ(t)), where y is some quantity of interest, t is time, μ is the deterministic mean response, and N is some distribution of errors with SD σ (e.g. normal). Even if one does not make such an assumption explicitly, when performing optimization for some objective function, such an assumption is always implicit. For example, using a least-squares objective function a normal distribution of errors is implicitly assumed. This, however, presents a problem when modeling strictly non-negative quantities such as organism densities. Since negative densities cannot be observed, the distribution of errors cannot be normally distributed. For large densities this is not really a problem in practice, but for densities close to zero the estimates will be clearly biased. This problem can normally be circumventing by studying the positive quantity on a log scale, but since the authors have several data points that are identically zero, this approach cannot be taken. Once again, I do not believe that this problem merits changing the model, but it I do think it merits some discussion by the authors justifying their approach.

Major comment 2: My second major concern is that the description of how the fitting procedure was carried out is insufficient. I glanced at the PotterWheel Matlab toolbox the authors used to fit their model, and it appears to have many different options for fitting models. The authors should provide sufficient description of their process for replication to be possible. As I understand that such a description might be rather lengthy and technical I think it would be fine to put this information in an appendix or in the supplementary materials.

I also have a specific question related to the fitting process. As far as I could understand from the documentation, the PotterWheel toolbox optimizes the objective χ^2 = d1^2/σ1^2 + ... + dn^2/σn^2 for data points 1 through n, where d_i is the distance of the fit from the data point i. However, several of the standard errors (especially towards the end of many time series) are in fact zero, which would imply an infinite χ^2. Could the authors clarify this point?

## Minor and line-by-line comments ##

Line 90: The sentence starting on this line appears to be incomplete.

Line 147 (and others): When a subscript is not a mathematical symbol it should be set in upright font (here, 'slope').

Line 167: For the sake of replicability, version numbers for matlab and the toolbox should be reported.

Line 183: X^2 -> χ^2

Lines 205-207: I cannot understand this sentence. Consider rephrasing.

Line 217: Given the error bars, I am not sure the results necessarily supports the conclusion of unimodality for r and r_slope. Did the authors perform any statistical tests supporting this conclusion? If so, they are not reported.

Line 243: 'More complete depiction' -> 'A more complete depiction' or 'More complete depictions'

Line 283: '...time estimated as the inverse of the r' -> '...time estimated as 1/r'

Figure 2 figure text: What does "SE of the mean of each point mean" Is this referring to the sample SD among the replicates, or is it something else? Why is it interpolated between points?

Reviewer 4 ·

Basic reporting

no comment

Experimental design

no comment

Validity of the findings

no comment

Additional comments

This study uses a Didinium-Paramecium predator-prey system to infer how temperature affects its dynamics. While the experimental data shows a clear trend in shifting cycles with warming, the main focus is uncovering the mechanisms leading to this pattern. This is an import and timely issue since, beyond individual increase of biological rates, these processes are not fully understood yet. An ODE-based predator-prey model is fitted to the observed time-series, allowing the direct estimation of all parameters involved. The presentation of the methods and the results is clear and straightforward. I really appreciate the effort and the complexity of confronting a continuous-time dynamical model with experimental time-series data. I have one concern about the methodology and hope that the authors carefully take it under consideration.

My main concern with their approach is the identifiability of the model. It includes 8 parameters describing a variety of processes: growth + density dependence, feeding interaction, interference, mortality + density dependence. All of them have to be informed by just 1 cycle in the dynamics. I understand that a flexible model is required to fit the data well, but that does not automatically imply identifiability of all parameters. Due to (1) process or observation stochasticity, (2) parameter redundancy in this highly nonlinear context, or (3) a general mismatch between model and reality, parameter estimates could be biased. Even if (3) could be ruled out by theory, with (1) and (2) the question remains, how can we trust the estimates.

The authors cite a previous simulation study (DeLong et al 2018), that tested model identifiability by fitting data which was generated with known parametrisation. However, that study used a simpler model with 4 parameters r,a,e,d (or 5, if carrying capacity K was included in the validation, I could not figure that out at first glance). I recommend that the authors include such a simulation study with their full 8 parameter model. Thus, doubts about biased parameters would be either ruled out, or, found bias or uncertainty in estimates could be included in the interpretation of their results. I believe that the paper would greatly benefit from such a validation.

other comments:

l. 190: “but these populations did not achieve and identifiable carrying capacity at all temperatures”. Are the observed densities in the range of these with predator treatments? If it is already difficult to fit carrying capacity in control treatments, I guess this holds for the full model under predation, too. Again, this can be tested in the proposed simulation study.

l. 216: “all of the parameters governing these equations changed with temperature”. Please be careful when comparing parameter estimates without a statistical test for difference. E.g., the variation in parameter e in Fig. 3F could be just random. Especially when the quantified uncertainty is just 1 sigma (68.5% CI).

Fig. 4: I find this figure hard to interpret. Maybe separate plots for pred and prey densities would make it more readable (as in Fig 2).

---

## Round 0.3 · Minor Revisions

Many thanks for the thorough revision. The manuscript was resubmitted to one of the previous reviewers who is satisfied with the changes, with one minor further suggestion.

Reviewer 4 ·

Basic reporting

no comment

Experimental design

no comment

Validity of the findings

no comment

Additional comments

The authors reduced the number of free parameters in the model fitting from 8 to 6 by carefully fixing two parameters to reasonable values. They further added a test for model identifiability of this 6-parameter-model by using simulated data and checking if „true“ input parameters can be recovered by the fitting routine.

Yet, I as far as I understand from the Matlab code, the simulated data does not contain any stochasticity (e.g., observation error). If this is the case, it is no surprise that the model fitting will return the input parameters exactly (Fig. S1) and the fitted trajectory perfectly matches simulated data. In a strict sense, this does not indicate model identifiability, it merely shows that the optimizer correctly converges to the minimum of the deviance function. Model identifiability would require that, even including some noise, this minimum is still close to the true input parameters.

I suggest that a little observation error is added to the simulated data to test this. Maybe this does not require another round of reviews if the results don‘t change. But it would rule out any concerns about biased or redundant parameters and non-identifiability.

---

## Round 0.4 · accepted · Accept

Many thanks for this final revision. I do not have anything more to add, besides congratulating you on acceptance of the manuscript!